# Direct observation of picosecond melting and disintegration of metallic nanoparticles

Yungok Ihm [1], Do Hyung Cho[2], Daeho Sung[2], Daewoong Nam [2], Chulho Jung[2], Takahiro Sato[3,4], Sangsoo Kim[5], Jaehyun Park[5], Sunam Kim[5], Marcus Gallagher-Jones [6,7], Yoonhee Kim[8,9], Rui Xu[6], Shigeki Owada[3], Ji Hoon Shim[1], Kensuke Tono[10], Makina Yabashi [3], Tetsuya Ishikawa[3], Jianwei Miao[6], Do Young Noh[8] & Changyong Song[2]

Despite more than a century of study, the fundamental mechanisms behind solid melting remain elusive at the nanoscale. Ultrafast phenomena in materials irradiated by intense femtosecond laser pulses have revived the interest in unveiling the puzzling processes of melting transitions. However, direct experimental validation of various microscopic models is limited due to the difficulty of imaging the internal structures of materials undergoing ultrafast and irreversible transitions. Here we overcome this challenge through time-resolved single-shot diffractive imaging using X-ray free electron laser pulses. Images of single Au nanoparticles show heterogeneous melting at the surface followed by density fluctuation deep inside the particle, which is directionally correlated to the polarization of the pumping laser. Observation of this directionality links the non-thermal electronic excitation to the thermal lattice melting, which is further verified by molecular dynamics simulations. This work provides direct evidence to the understanding of irreversible melting with an unprecedented spatiotemporal resolution.

[1] Department of Chemistry, Pohang University of Science and Technology, Pohang 37673, Korea. [2] Department of Physics, Pohang University of Science and Technology, Pohang 37673, Korea. [3] RIKEN SPring-8 Center, 1-1-1 Kouto, Sayo, Hyogo 679-5148, Japan. [4] Linac Coherent Light Source, SLAC National Accelerator Laboratory, Menlo Park, CA 94025, USA. [5] Pohang Accelerator Laboratory, Pohang 37673, Korea. [6] Department of Physics & Astronomy and California NanoSystems Institute, University of California, Los Angeles, CA 90095, USA. [7] Department of Chemistry and Biochemistry, University of California, Los Angeles, CA 90095, USA. [8] Department of Physics and Photon Science, Gwangju Institute of Science and Technology, Gwangju 61005, Korea. [9] European XFEL GmbH, 22869 Schenefeld, Germany. [10] Japan Synchrotron Radiation Research Institute, 1-1-1 Kouto, Sayo, Hyogo 679-5198, Japan. Correspondence and requests for materials should be addressed to C.S. (email: cysong@postech.ac.kr)

Ultrashort lasers provide mode-selective access to entangled physical processes. Femtosecond laser irradiation of nanoparticles, by triggering electron excitation exclusively, provides a rich platform for observing broad physical phenomena ranging from material phase changes to the generation of ultrashort pulse radiation, which have attracted keen interest[1–8]. Direct visualization of ultrafast nanoparticle melting and disintegration has been sought, in particular, to gain an *ab initio* insight into the century old question of solid melting, as the first step toward the understanding of material deformation phenomena[9–14]. Until now, however, this effort has been limited to model dependent interpretation or weakly perturbed, reversible processes[15–17].

In this work, we perform time-resolved X-ray free electron laser (XFEL) single-pulse imaging experiments on femtosecond (fs) laser-irradiated nanoparticles in highly nonequilibrium states, and unveil the atomic process involved in irreversible melting and disintegration by providing real images of specimens at better than 10 nm and 10 ps spatiotemporal resolution.

## Results

**Femtosecond X-ray single-pulse pump-probe imaging experiments.** XFEL single-pulse pump-probe imaging experiments were carried out at SPring-8 Angstrom Compact X-ray free electron Laser (SACLA)[18]. Femtosecond IR laser (800 nm wavelength and 50 fs pulse-width) was used as a pumping source (Fig. 1 & Methods). Using X-ray pulses, we took femtoseconds snapshot pictures of individual Au nanoparticles of 100 nm in diameter irradiated by single IR laser pulses[19]. Incident optical laser fluence was 870 mJ cm$^{-2}$ and the effective absorbed energy density was estimated to be ~11 kJ cm$^{-3}$, which provides enough energy to raise the lattice temperature above the melting point (Supplementary Note 1)[20].

**Picosecond and nanoscale observation of irreversible transitions.** XFEL single-pulse diffraction imaging has revealed the irreversible melting and disintegration process of single Au nanoparticles irradiated by single fs IR laser pulses. With the progress of material deformation, single-pulse diffraction patterns show anisotropic distortion from Airy patterns as well as overall shrinkage of speckle sizes (Fig. 2). This indicates that the melting of Au nanospheres proceeds with a local deformation in addition to volume expansion. In particular, the distortion proceeds with

notable decrease in the speckle visibility along the laser polarization direction, which implies the loss of sample integrity along that direction (Fig. 2c). This feature is more clearly distinguished with the line plots (Fig. 2e). The line plots at 80 ps delay can be compared with those from intact samples displaying stronger smearing of the fringe oscillations along the polarization direction (Fig. 2e). After 140 ps, speckles in the diffraction patterns disappeared and anisotropic, butterfly-shaped incoherent diffraction patterns were observed, indicating the complete destruction of the nanoparticles (Fig. 2d). The direction of the 'butterfly-wing' dif-

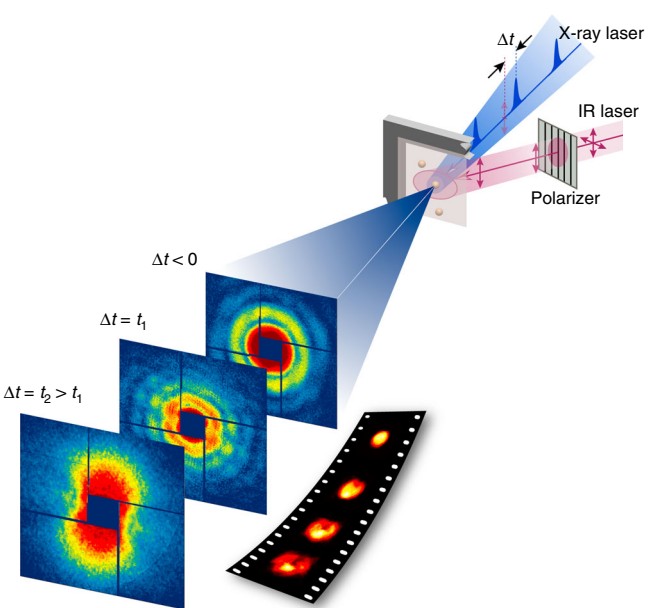

**Fig. 1** Single-shot pump-probe imaging using femtosecond X-ray laser pulses. A femtosecond IR laser irradiates Au nanoparticles, driving the melting process, and single-pulse XFEL diffraction patterns are collected after planned delays to track the reaction on the picosecond time scale. Single-shot images of nanoparticles undergoing an irreversible melting transition are reconstructed from the diffraction patterns to unveil the ultrafast melting process directly at nanoscale resolution

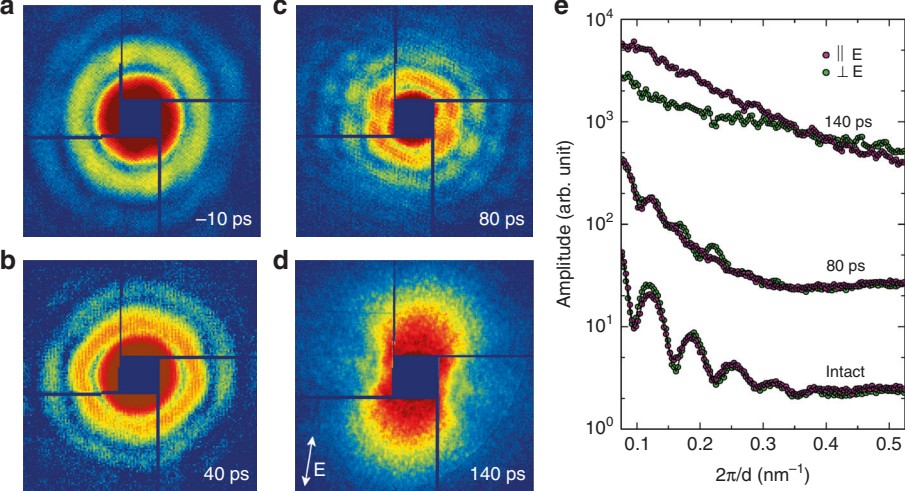

**Fig. 2** Time resolved single-shot speckle patterns using femtosecond X-ray laser. **a–d** Single-shot coherent diffraction patterns from single Au nanoparticles are displayed showing anisotropic distortion in the speckle pattern with the melting in progress. **e** Line plots along the directions parallel and perpendicular to the laser polarization are compared. On melting, the fringe oscillation smears out more strongly along the polarization direction (80 ps) and eventually higher diffuse scattering signal is detected on complete melting at 140 ps (**e**). The polarization direction of the IR pump laser is shown with the arrow in **d**

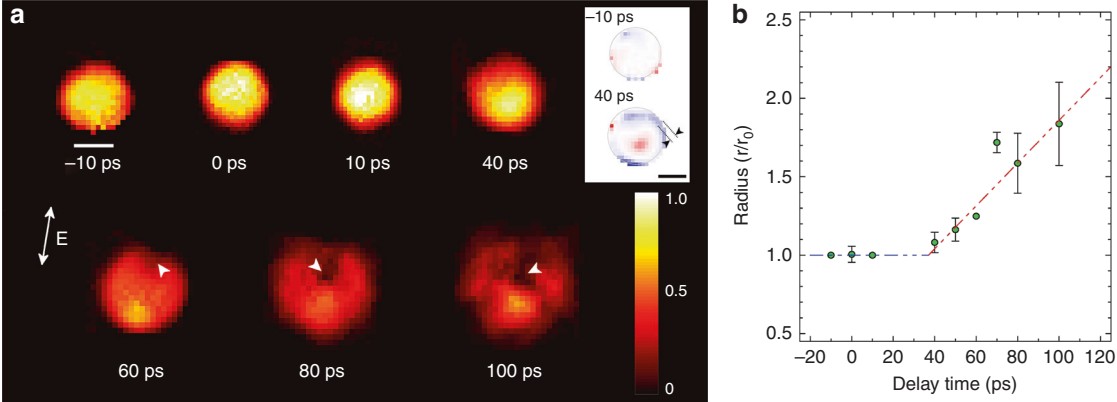

**Fig. 3** Time resolved single-shot images of single Au nanospheres during melting. **a** Single-shot images visualize the melting processes in single Au nanospheres through the projected density (**a**). White arrowheads in 60–100 ps in **a** track the void formation. The variation of the projected density for 40 ps near the surface is emphasized by displaying the density difference from ideal sphere, which visualizes inhomogeneous melting at the surface with layer thickness of ~10 nm marked with two arrowheads in (inset). The scale bar shows 50 nm and the polarization direction of the pumping laser is shown with the arrow in **a**. The scale bar for the colormap shows scaled projected density to have one as the maximum value of the original intact sphere. **b** The temporal evolution of the size of the Au nanosphere on the melting is obtained. More rapid expansion of the particle size during the melting transition is noted from ~ 40 ps as from with the temporal evolution of sample radius in **b**. Error bars indicate the standard deviations. Lines are guides to the eyes and its slope is ~700 m s$^{-1}$

fraction pattern is along the polarization direction, which is consistent with the diffuse scattering patterns resulted from ablated nanoparticles after nanoseconds delay[21]. This experimental observation implies that a simple radial expansion or two-layer model with dense-core and lower-density shell is insufficient to account for the melting process[15].

By retrieving the phases of the measured single-pulse diffraction patterns, we have reconstructed images representing 2D projections of the electron density of spherical nanoparticles (Fig. 3, Methods, Supplementary Figs. 1–4 and Supplementary Note 2)[22]. All the images were obtained from individual nanoparticles with very consistent 3D morphology, each of which was exposed to the fs IR laser only once (Methods). The reconstructed images were sorted out for the same delay time and tens of images were collected for each delay time (Supplementary Fig. 2). A specimen rotated 180 degree along the incident X-ray beam axis results in the same X-ray diffraction pattern as the unrotated one, and there is no physical distinction between the upper and lower hemisphere in this experiment. We, thus, have aligned the images to display the void appearing in the upper hemisphere by rotating 180 degree, if in need, along the axis normal to the image plane without losing generality.

From the collection of images, we chose one representing the average behavior for each delay time. The melting evolution of spherical Au nanoparticle with the increased degree of inhomogeneous density variation is shown as a function of the delay time (Fig. 3a & Supplementary Movie 1). The images were displayed with the same colormap scale to make a direct comparison of the density variation relative to the intact nanoparticle. Images at −10 and 0 ps show no discernible density fluctuations, which is expected for intact nanoparticles (Fig. 3a). Images taken after the IR laser pulse show that the melting and disintegration proceed with complex multistep rearrangements of atomic distribution (Fig. 3a). The process started with a density reduction from the surface as frequently noted in the heterogeneous melting. Reduced density at the surface was observed for images at 10 ps and more clearly at 40 ps. To better visualize the density variation near the surface, the image at 40 ps is compared to an ideal sphere of the same volume with homogeneous density distribution. The difference in the projected densities is shown in the inset (Fig. 3a). The blue colored region near the surface represents the decrease of

the projected density relative to an ideal sphere, showing the surface melting at 40 ps.

The projected density decreased as the nanoparticle started to melt and expand (Fig. 3). In particular, as the melting continued, the ion-dispersed regions were developed inside the sphere, which were further enhanced to become a void (marked as arrowheads in Fig. 3a). While penetrating toward the sphere center, the void expanded with the increase of the density variation, which eventually resulted in a complete destruction of the nanoparticle. Using the collected images for each delay time, we estimated the size of spherical Au nanoparticles during the melting and `disintegration transition (Fig. 3b). The expansion of the nanoparticles became pronounced after 40 ps, and they were completely disintegrated around 100 ps. The radial expansion rate was estimated to be ~700 m s$^{-1}$, which is slightly slower than the speed of sound.

Careful inspection of the images reveals that those ion-dispersed and concentrated areas were formed along the polarization direction of the IR laser (Fig. 3 and Supplementary Fig. 2). This was further corroborated by the experiment that after flipping the polarization axis of the IR laser pulse, the ion density distribution followed the newly defined polarization direction (Supplementary Figs. 3 and 4). Given the spherical morphology of the Au nanoparticles, this anisotropy in the ion redistribution is an induced feature controlled exclusively by the pump laser. Stronger electron ionization along the polarization direction is expected from the dipole transitions[23,24]. However, the direct influence of this anisotropic electron excitation occurring on the femtosecond time scales to drive directional rearrangement of atoms in several tens of ps time span was unveiled through direct nanoscale images as a function of the delay time, which to our knowledge has not been previously reported. We anticipate that the explicit observation of how the laser excited electrons induce ionic displacements at different delay times will extend our understanding on the electron-induced dynamics of ions beyond the Born-Oppenheimer adiabatic regime[25].

**Two-temperature molecular dynamics simulations**. To gain physical insights into the melting process at the atomic scale, we have compared the experimental results with the atomistic molecular dynamics (MD) simulations. A two-temperature model

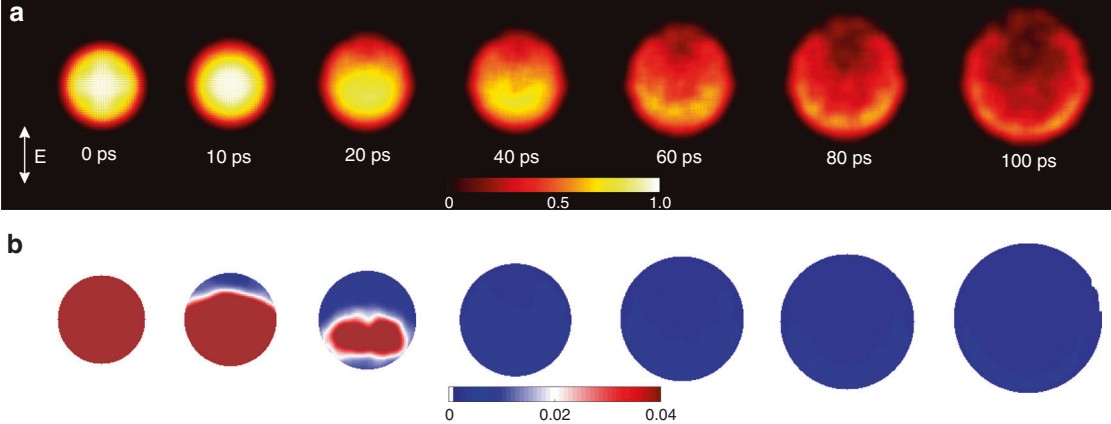

**Fig. 4** Two-temperature molecular dynamics simulated atomic density and disordering. **a** Projected atomic density was obtained showing anisotropic distribution. The Au nanoparticle is distorted from the intact spherical morphology upon melting with a preferential shift of high-density region along the electric field direction, similar to the experimental observation. Images are drawn with the same color map scale shown with higher density in bright yellow color. **b** The order parameters are obtained displaying the anisotropic progress of disordering from the crystalline phase on melting. The area with dark brown color indicates the region in a solid phase. The color map shows the numerical value of the order parameters, whose value smaller than 0.04 is regarded as disordered states

was implemented into the MD simulations to account for the irreversible melting of the Au nanospheres (Methods and Supplementary Note 3)[26]. In the two-temperature molecular dynamics (TTMD) simulations, Au atoms absorbed the energy from oscillating electron clouds formed by fs laser ionization with asymmetric field enhancements, which is consistent with the observed asymmetric melting along the polarization direction.

Projected density maps of ions obtained from the TTMD simulations show good agreement with experimental results (Figs. 3a and 4a & Supplementary Movie 2). To visualize development process at the atomic scale, we calculated the order parameter using the displaced atomic positions from the TTMD (Fig. 4b & Supplementary Note 4)[27]. A perfect crystal has an order parameter of one and any value smaller than 0.04 is defined as the loss of crystalline order[27]. It shows that the asymmetric disordering starts from a pole region at the surface as early as ~10 ps to propagate over the whole nanosphere by ~ 40 ps, after which the void develops (Fig. 4b & Supplementary Movie 3). Further detailed descriptions on the TTMD and the order parameter calculations together with interpretation are provided (Supplementary Discussion).

## Discussion

The void results from the release of the enhanced local pressure due to the increase of atomic displacements more amplified along the polarization direction. The fs laser pulse excites localized surface plasmon, which induces enhanced electric fields at the nanoparticle surface and interior[28, 29]. This near-field enhancement is intense near the pole region of the sphere surface and decreased with radial distance inside the nanoparticle[28, 30]. As a result, a local area with more energetic electrons (i.e., a hot spot) is formed instantaneously, which eventually thermalizes and equilibrates with other electrons[31]. These transiently excited hot electrons create the reduced charge screening to weaken the interatomic bonding between Au ions in the local area[32]. Several mechanisms can account for the reduced screening, including the excitation of conduction electrons, photoelectron emission via single or multiphoton absorption, etc[33]. The combination of the anisotropically weakened interatomic bonding and the energy transfer from the highly excited electrons to the lattice trigger ionic pressure accumulation, which is relieved primarily through the region with the weakened interatomic bonding. This process

drives the anisotropic melting with the void formation as directly observed from our single-shot imaging experiment[20, 34–37].

We have applied XFEL single-pulse pump-probe imaging to unveil the dynamic processes underpinning the irreversible melting of Au nanoparticles. Images directly visualizing the ion distribution were obtained at sub-10 nm spatial and 10 ps temporal resolution. Anisotropic redistribution of ions in response to the electric field direction of femtosecond IR laser pulses was observed. By explicitly demonstrating the impact of femtosecond electron excitation on the nanoparticle disintegration process, this experimental observation unveiled the direct link between non-thermal electronic excitation and thermal lattice melting process, which was corroborated by atomistic TTMD simulations. We believe that the ability to image ultrafast melting at the nanoscale and elucidate the interrelation between ultrafast dynamics of electrons and ionic motions provides an important step towards resolving the century long mystery of how the solid melting phenomena occur.

## Methods

**Single-pulse pump-probe imaging experiment and data acquisition**. Ultrafast single-pulse pump-probe imaging experiments were carried out using the Multiple-Application X-ray Imaging Chamber (MAXIC) installed at EH3 on the BL3 of SACLA[38]. The X-ray energy was fixed to 5.5 keV and the average pulse energy at the sample position is ~100 μJ with pulse-to-pulse fluctuation within 10%. A pair of K–B mirrors has focused single X-ray pulses to 1.5 μm (H) × 1.5 μm (V) spot at the sample position. A single X-ray pulse delivering ~$10^{10}$ W μm$^{-2}$ leaves a burned hole on the membrane after an exposure. The near IR (λ = 800 nm) Ti:sapphire laser of 50 fs pulse-width was focused to a 50 μm spot in diameter. The polarization of the IR pump laser was controlled using a half-wave plate. XFEL single-pulse diffraction experiments were carried out while scanning the membrane with the focused X-ray beam with the intersection of the IR pump laser (Fig. 1). Coherent diffraction patterns were recorded on an octal sensor multi-port charge-coupled device (MPCCD) detector having a 2048 × 2048 array of 50 μm pixels. The sample to detector distance was kept at 2.2 m.

With a tilt angle between the pump laser beam and sample plane of ~30 degree, the spot size on the membrane is elongated slightly (<140 %) along the horizontal direction. The incident laser fluence is ~870 mJ cm$^{-2}$ (Supplementary Note 1)[20]. The spatial overlap of the X-ray laser with the IR pump laser's footprint at the sample plane was monitored using inline optical microscopes. The delay time was controlled by an optical trigger at better than a few ps resolution.

**Fixed target sample preparation for XFEL single-pulse imaging**. Unconjugated spherical Au nanoparticles of 100 nm diameter (Nanopartz$^{TM}$) were mounted on 100 nm thin $Si_3N_4$ membranes with an array of multiple windows (35 by 35 arrays of 200 × 200 μm windows), custom designed (manufactured by Silson LTD) for single-pulse diffraction experiments with the fixed target sample delivery scheme[39]. Au nanoparticles in deionized purified water were spin-coated on the $Si_3N_4$

membranes with a nominal inter-particle distance of 3–5 μm to accommodate maximum single-particle hit rate. Single-pulse imaging experiments, in the absence of pump laser, were performed by raster scanning of the sample stages to keep the shot-to-shot distance longer than 25 μm at 10 Hz XFEL operation condition without pump laser. With the pump laser on, the raster-scan mode was changed to accept one pulse for each window to ensure hitting the particle only once by the pumping laser. Once the $Si_3N_4$ window is exposed to an IR laser pulse, it becomes broken completely. Laser exposure of the nanoparticles sitting nearby windows is prevented by separating the window to window distance far more distant than laser footprint. The footprint of the optical pumping laser is less than 100 μm, whilst the window-to-window distance is 380 μm. This special protocol is applied to make sure that nanoparticles are hit only once by the IR laser as well as XFEL pulses.

**Single-particle diffraction patterns, phase retrieval, and single-shot images.** Coherent diffraction patterns from single Au nanoparticles are obtained by exposing fresh nanoparticles to single pulses of the femtosecond XFEL. For the data analysis, we discarded the diffraction patterns of multiple-particle hits displaying interference fringes or speckles with much finer oscillation periods than expected from a single particle, and only used the data from single-particle hits.

Each single-shot coherent diffraction pattern is phase retrieved to acquire an image of each independent nanoparticle following the well-established coherent diffraction imaging procedures[40–42]. This phase retrieval of acquiring a real image is not a fit process but an optimization approach to the inverse problem through numerical iterations following a phasing algorithm[43]. Coherent diffraction patterns were measured with a sampling interval finer than the Nyquist frequency determined by the inverse sample size is used for amplitude modulus of the Fourier transformation of the specimen[44, 45]. The phases of the diffraction patterns are not measured explicitly, but become retrieved from such oversampled coherent diffraction patterns.

The phase retrieval is performed staring with the randomly guessed phase for the first step. The measured amplitude modulus with this random phase is inverse Fourier transformed to render an image in real space. This image, which is usually far from an actual image of the specimen at this initially stage, is further inspected to comply with the rationally imposed constraints such as the extent of the image, positive value in the image density, etc. This constrained image is numerically Fourier transformed to obtain a diffraction pattern with the amplitude modulus and phase. In this stage, the amplitude is replaced with the experimentally measured one while keeping the phase and inverse Fourier transformed to get an image again. Obtained image is constrained again and Fourier transformed back for the diffraction pattern. This process is repeated back and forth and finally converges to render an image of real specimen. As noted in this phase retrieval process, all the measured data are directly and unbiasedly used for imaging, which is clearly distinguished from data fitting process used elsewhere. Detailed descriptions on the phase retrieval process and the algorithm employed are provided (Supplementary Note 2).

Data collections are repeated for each delay time and multiple numbers of images are for the same delay time (Supplementary Figs. 2 and 4). An image well delivering the average behavior of the images of the same delay time is chosen to represent the image of the given delay time.

**Atomistic two-temperature molecular dynamics simulations.** TTMD simulations were carried out with an unsupported 10 nm nanosphere made of Au atoms arranged in the face centered cubic crystal symmetry. An embedded-atom-method (EAM) and the MD time step of 0.7 fs were used. The nanosphere was first equilibrated at 300 K for 2 ps. The nonequilibrium states of the electronic and ionic systems of the Au nanosphere melting upon ultrashort laser irradiation were described by combining the atomistic molecular dynamics with the two-temperature model (see Supplementary Note 3 for the details of the TTMD method). The incident laser fluence and the pulse duration were 870 mJ cm$^{-2}$ and 50 fs. The TTMD simulations were performed on spherical grids with energy absorption from the oscillating plasma wave with isotropic and anisotropic field enhancements (Supplementary Note 3).

## Data availability
The data that support the findings of this study are available from the corresponding author upon reasonable request.

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

## Acknowledgements

The XFEL experiments at SACLA is approved by Japan Synchrotron Radiation Research Institute. We appreciate stimulating discussion with K.B. Lee and the supply of patterned specimens from Y. Hwu and K. Liang. Authors are grateful to A. Suzuki, SACLA data acquisition and engineering team for technical support on experiments at SACLA. Y.I. acknowledges J.R. Morris for helpful discussions and providing a classical MD code. This work was supported by the National Research Foundation (NRF) of Korea (Grant No. 2016R1A2B3010980, 2015R1A5A1009962, 2017K1A3A7A09016380), and in part by NRF (No. 2017R1D1A1B03032069, 2018R1D1A1B07040727). J.M. acknowledges the support by STROBE: A National Science Foundation Science & Technology Center (Grant No. DMR 1548924).

## Author contributions

C.S. conceived the project. Y.I. in communication with J.S. performed the TTMD simulations. C.S., D.C., C.J., and Y.I. analyzed the data. D.C., D.S., D.-W.N., C.J., T.S., S.-S.K., J.P., S.-N.K., M.G.-J., Y.K, R.X., S.O., K.T., M.Y., T.I., J.M., D.-Y.N., and C.S. contributed to the experiment. C.S. and Y.I. wrote the manuscript with the inputs from all authors.

## Additional information

**Competing interests:** The authors declare no competing interests.

