## [Peer Review File · Nature Communications]

Editorial Note: Parts of this peer review file have been redacted as indicated to remove third-party material where no permission to publish could be obtained.

Reviewers' comments:

Reviewer #1 (Remarks to the Author):

This manuscript is clearly written and tells an interesting story of the processes involved in the laser driven explosion of 100 nm Au nanoparticles spin coated onto a membrane. The story may be true, given the number of steps involved in the simulation, but there is little evidence for many those assumptions.

As it stands, the work is unsuitable for publication. It is too speculative and need detailed statistics about how often the particular experimental results were obtained. The authors are free to speculate with modeling the effects seen, but the experimental results (and how much they vary) have to be accounted for in full detail.

How were the data sorted to ensure that a single particle was in the beam in each shot on a fresh membrane? If multiple particles were exposed, how were the coherent interference fringes between them removed?

The title is totally inadequate. Because of entropy release, all melting should be irreversible. It can only be "reversible" on very short time scales, such as probed with an XFEL, but not in the general case. I would insist on a more specific title, perhaps about the polarization effects.

The work is motivated towards "understanding material deformation phenomena", which is normally thought to be the domain of elasticity/plasticity in materials science. I do not think this is an appropriate reason to report the results found, or to cite refs 17 and 18.

How many times were the experimental results reproduced, for example the one shown in Fig 2d?

I am troubled how a symmetric diffraction pattern (Fig 2d has very close to mm symmetry) inverts to a highly asymmetric image of a particle with a crater ("void") on one side and most of the density on the other. One mirror is preserved in the image but not the other. How unique are the images, given that "The fits were visually inspected individually to ascertain faithful fits."? How was the inversion constrained? There are almost no details of the phasing calculation provided in the manuscript or the SI and this is central to the experimental story.

What is the difference between to the top and bottom row of images in SI Figs 2 and 4?

I have similar doubts about the simulations. How is symmetry broken there, given that all input assumptions are symmetric.

line 350 Density difference map? The text says "normalized density variation from a calculated density for an ideal sphere", but is the ideal sphere enlarged to account for the expansion of the image? There would be no logic for assuming the material should retain a spherical shape as it expands, so this construction is a rather weak idea.

Reviewer #2 (Remarks to the Author):

The authors present combined X-ray scattering experiment and TMM-MD simulations of a single spherical gold particle irradiated by a near-IR laser pulse to induce melting and particle destruction.

The single-shot experiment with ultrashort X-ray pulses at an X-FEL source seems to be heroic and the iterative reconstruction algorithm apparently producing interesting results.

After a proper discussion this result may be publishable, although not necessarily in a leading journal. But first a complete revision needs to take place.

1. The most important point of criticism is that the authors fail to correctly describe their experiment and put it into context of current knowledge on laser-matter interaction in metal nanoparticles. It should be clearly stated that

a) the described experiment is a SAXS experiment and as such can not comment at all on crystallinity of the sample. The density change from solid to liquid is probably much too small to be distinguished.

b) The given fluence value seems odd. Fluence thresholds for particle modifications are frequently studied and have shown that even in a dense medium (such as a liquid) with stronger plasmon resonance about 10-15 mJ/cm² are needed to bring the particles close to the melting point (Boutopoulos, *Nanoscale* 2015, Plech *Phys. Rev. B* 2004, Katayama *Langmuir* 2014, Lombard, *Phys. Rev. Lett.* 2015, Hashimoto *J. Photochem. Photobiol. Rev.* 2012). Melting enthalpy has to be added here. Even if the given fluence is supposed to melt the particles completely it does not necessarily result in particle disintegration. Nevertheless, near-field induced ablation has been reported earlier to happen around the melting point, not necessarily in the gold liquid phase (ref. 22, which btw excludes the notion "for the first time" on top of page 5). So a meaningful report needs to prove the real processes and thresholds.

2. A close inspection of the reconstructed projected density maps reveals some odd perceptance:

a) I assume that all data stem from single-shot excitation of individual particles (i. e. every particle can only be used once before both particle and support are destroyed). Can the authors confirm this? In that case it seems surprising that for instance delays of 60, 80 and 100 ps produce very similar damage patterns, which imply a time sequence on the very same particle. Did the authors sort a large number of data in order to pick particularly those, which seem similar?

b) the difference map in fig. 3. b is strange. Apparently in some spots the intensity increases by 70 %. How can that happen? Is the average value floating?

3. the discussion of a mechanistic excitation cascade on page 6 is clumsy and not precise. An assumption that momentum transfer from fast electrons to the ions might lead to anisotropic ion movement is pure speculation. Different views of local field effects, such as from emitted photoelectrons along the plasmon resonance is not discussed. Anyway, the simulation cannot give an answer to that as it only artificially imposes an asymmetry in energy deposition in eq. S4, not backed by any ab-initio knowledge.

Also: "The void results from an enhanced local pressure caused by increased atomic displacements more amplified along the polarization direction." seems strange. Shouldn't it be a drop in pressure in the center that allows creating a void as usual in tensile disruption? Anyway, at this point comments on strain do not seem to be backed up by underlying (simulation) data, but rather serving as heuristic description.

4. Besides the nice data it is difficult to distill real new insight gained by the study by just watching single events. I guess, figure S4 would be the most quantitative outcome, worth being incorporated in the main text (while the dotted lines are only guide to the eye?)

Response to reviewers' comments

We appreciate the referees' careful reading of our manuscript and their invaluable comments. We have replied to all the raised comments as detailed below; the manuscript is revised fully implementing all of the referees' suggestions. Our point-to-point replies to referees' comments are as follows.

[Reviewer #1]

This manuscript is clearly written and tells an interesting story of the processes involved in the laser driven explosion of 100 nm Au nanoparticles spin coated onto a membrane. The story may be true, given the number of steps involved in the simulation, but there is little evidence for many those assumptions. As it stands, the work is unsuitable for publication. It is too speculative and need detailed statistics about how often the particular experimental results were obtained. The authors are free to speculate with modeling the effects seen, but the experimental results (and how much they vary) have to be accounted for in full detail.

[Comment 1]

How were the data sorted to ensure that a single particle was in the beam in each shot on a fresh membrane? If multiple particles were exposed, how were the coherent interference fringes between them removed?

<Author reply>

To accommodate better understanding on the presented research for researchers with diverse disciplines, we have extended the description of the data collection and analysis in the manuscript. In the methods section we have included a new section on "Single particle diffraction patterns, phase retrieval and single-shot images" in pages 9-11 of the revised manuscript.

We used multiple window Si_3N_4 membranes for the single shot experiments. After each single-pulse exposure, the membrane of the exposed window is destroyed and we do not (cannot) take a second shot from the same window. Furthermore, the size of the IR laser footprint on the sample position is kept much smaller than the size of the window, which also ensures that the optical laser is not extended to expose particles on other windows nearby. Abiding by these precautions, we take single-pulse X-ray diffraction patterns from fresh specimens only. The possibility of including data from multiple-particle hits is ruled out completely during the data analysis step. From the collected single-pulse diffraction patterns, we have selected only patterns that correspond to single particles being illuminated by an XFEL pulse by discarding the patterns displaying interference fringes within speckles or speckleless-

features resulting from multiple-particle illumination. To facilitate understanding, we provide an image of the membrane broken by the IR laser pulse and diffraction patterns of multiple particle hits collected during the experiment, which are excluded in the data analysis made here. One can note the additional fringes in these patterns which clearly distinguish them from single-particle diffraction patterns.

Images above show the SiN membrane used for the experiment. One membrane is made of 33 x 34 arrays of SiN windows, which are almost transparent to the XFEL pulses. The single-shot pump-probe experiment is performed such that only a single IR laser pulse is exposed to each fresh window (left), and the window becomes broken eventually (right). One can note that any repeated hitting of the same particle by a laser pulse is not feasible.

Figures below display diffraction patterns from multiple-particle hits. Patterns show interference fringes, which are the clear indication of the multiple particle hit to guide the data analysis. Corresponding phase retrieval images from the diffraction patterns are included as insets.

[Comment 2]

The title is totally inadequate. Because of entropy release, all melting should be irreversible. It can only be "reversible" on very short time scales, such as probed with an XFEL, but not in the general case. I would insist on a more specific title, perhaps about the polarization effects.

<Author reply>

In the revision, we have changed the title as “Direct observation of ultrafast melting and disintegration of metallic nanoparticles using XFEL single-pulse imaging”.

On the other hand, we want to clarify one thing. The title was chosen to emphasize the capability of our XFEL single-shot imaging technique to successfully image ultrafast irreversible processes, which has not been demonstrated before at this high spatio-temporal resolution. With this new development, the probe can successfully trace the complete particle deformation process at a resolution far exceeding any probe available thus far. The title was not intended to mean that the irreversibility of the observed melting is the novelty. The capability to observe ultrafast irreversible processes at the highest spatio-temporal resolution is the novelty of our work.

The most notable contribution of our work is the experimental realization of the observation of irreversible transformations during the melting and disintegration of metallic nanoparticles within a very short time frame (~ tens of picoseconds) and at sub-10 nm spatial resolution. High resolution imaging of such ultrafast phenomena is beyond the capability of existing techniques. Nearly all high-resolution imaging techniques thus far have been limited to imaging reversible processes through a stroboscopic probing scheme. The title of our manuscript was to emphasize this newly demonstrated experimental capability that pushes the limit of spatio-temporal resolution of nanoscale single-shot imaging. Nevertheless, in order to clarify our purpose while clearing out any potential confusion, we have changed the title to “Direct observation of ultrafast irreversible melting and disintegration of metallic nanoparticles using single-pulse XFEL diffraction imaging” in the revised manuscript.

[Comment 3]

The work is motivated towards "understanding material deformation phenomena", which is normally thought to be the domain of elasticity/plasticity in materials science. I do not think this is an appropriate reason to report the results found, or to cite refs 17 and 18.

<Author reply>

We have removed the references 17 and 18 in the revised manuscript.

[Comment 4]

How many times were the experimental results reproduced, for example the one shown in Fig 2d?

<Author reply>

We have repeated the whole experiment multiple times, and for each delay time several images were reconstructed, which are now provided in the revised SI. For each time delay about 10 images were recorded, as displayed in the Supplementary Information. We made a more detailed reply on this matter after comment 6. In short, the experiments have been repeated several times, and also other images obtained for the same experimental conditions are provided in SI Fig.2 & 4.

[Comment 5]

I am troubled how a symmetric diffraction pattern (Fig 2d has very close to mm symmetry) inverts to a highly asymmetric image of a particle with a crater ("void") on one side and most of the density on the other. One mirror is preserved in the image but not the other. How unique are the images, given that "The fits were visually inspected individually to ascertain faithful fits."? How was the inversion constrained? There are almost no details of the phasing calculation provided in the manuscript or the SI and this is central to the experimental story.

<Author reply>

In answering directly to the raised question on an asymmetric image from a symmetric diffraction pattern, we provide results from simple simulations below. The attached figures below show the calculated coherent diffraction patterns from the real images shown as insets. The calculated patterns exactly simulate the kind of data collected during our coherent diffraction experiments. These simulated diffraction patterns are directly calculated from the Fourier Transform of the image, which provides the answer to the raised question. Please note that an image without inversion symmetry still produces a diffraction pattern with inversion symmetry (left). The six-fold symmetry pattern (left) is clearly distinguished from other six-

fold patterns originating from different images containing six-fold symmetry, including inversion symmetry as questioned by the referee (center & right).

Furthermore, in order to assist the referee and researchers of broad expertise in understanding the experimental methods and the process to recover images from coherent diffraction patterns, we have largely expanded the description of the imaging and phase retrieval methods in the revised main manuscript starting with “All the images -- , -- the average behavior for each delay time.” in page 4 and also including new section in the Methods on “Single particle diffraction patterns, phase retrieval and single-shot images”.

In short, the phase retrieval is not accomplished from any fit to the measured data. The phase information is buried in the coherent diffraction pattern. Especially when the diffraction pattern is collected at a fine enough frequency such that the sampling frequency is more than twice finer than the Nyquist frequency of the inverse of size of the specimen known as the oversampling method proposed by D. Sayre in 1952 and first demonstrated by one of us (J. Miao, 1999), the phase information can be retrieved. Operationally, the phase information is retrieved via numerical iteration through phase retrieval algorithms such as hybrid-input-output method (J. Fienup 1979). There are various versions of phase retrieval algorithms, which are basically about the optimization methods in dealing with the inverse problem.

[Comment 6]

What is the difference between to the top and bottom row of images in SI Figs 2 and 4?

<Author reply>

Images on the top and bottom rows of the original SI Figs. 2 & 4 were from different Au nanoparticles taken under the same experimental conditions, delay time and laser polarization. They were shown to demonstrate that the images from other Au nanoparticles also exhibit similar behavior under the same experimental conditions. These images have been provided in the SI to show that our interpretation of the single particle imaging data is not from a single particular event but based on common features observed in other images taken under the same experimental conditions. In the revised SI, we now display the whole collection of images instead of these selective ones to avoid further confusion in Supplementary Figs. 2& 4. This also provides the answer to comment 4 raised earlier.

[Comment 7]

I have similar doubts about the simulations. How is symmetry broken there, given that all input assumptions are symmetric.

<Author reply>

The MD simulations were performed for an asymmetric energy input from the plasmon excitation as already detailed in the Supplementary Information. The experimental images clearly show broken symmetry along the laser polarization direction with void formation shifted towards one side. Justification to the simulation process can be found from other report as well. For instance, Varin and colleagues (Varin, C., Peltz, C., Brabec, T. & Fennel, T. Attosecond Plasma Wave Dynamics in Laser-Driven Cluster Nanoplasmas. *Phys. Rev. Lett.* **108**, 175007, (2012).) found that plasma waves excite electron hotspots and localized electric field fluctuations in the nanoparticle interior, which induce symmetry breaking in the course of nanoparticle melting and destruction. This report supports our model of asymmetric energy input with experimental results giving asymmetric ionic density evolution.

[Comment 8]

Line 350 Density difference map? The text says "normalized density variation from a calculated density for an ideal sphere", but is the ideal sphere enlarged to account for the expansion of the image? There would be no logic for assuming the material should retain a spherical shape as it expands, so this construction is a rather weak idea.

<Author reply>

The difference map was designed to emphasize local distortion of the Au particle from an ideal, i.e. non-distorted, spherical nanoparticle of the same mass. Certainly, there is no reason to assume that material should retain a spherical shape, but the assumption of ideal sphere is not critical for this analysis at all. The morphology of an ideal sphere provides reference densities for identifying regions within the local map displaying distorted density with density surplus or deficiency. The difference map is used to emphasize the density contrast within the nanoparticles.

Similar comments about mis-interpretation of the figure were raised by the second referee. As such, to prevent mis-interpretation of the result, we have now changed the figure to remove the independent panel displaying the difference map (Fig. 3). Other than better visualizing the subtle density variation at 40 ps, the difference map did not add any new information in the submitted version of the manuscript, and following the comments we removed the difference map display from the whole series of images. Further to accommodate better understanding on the actual projected density change, we revised the Fig. 3a to apply the same color map scale for the whole series of images now. This helps to gain an insight on the projected density variation directly compared to the original intact particle.

[Reviewer #2]

The authors present combined X-ray scattering experiment and TMM-MD simulations of a single spherical gold particle irradiated by a near-IR laser pulse to induce melting and particle destruction. The single-shot experiment with ultrashort X-ray pulses at an X-FEL source seems to be heroic and the iterative reconstruction algorithm apparently producing interesting results. After a proper discussion this result may be publishable, although not necessarily in a leading journal. But first a complete revision needs to take place.

[Comment 1]

1. The most important point of criticism is that the authors fail to correctly describe their experiment and put it into context of current knowledge on laser-matter interaction in metal nanoparticles. It should be clearly stated that

[Comment 1-a]

a) the described experiment is a SAXS experiment and as such can not comment at all on crystallinity of the sample. The density change from solid to liquid is probably much too small to be distinguished.

<Author reply>

Indeed, we do not distinguish the crystallinity directly from the image with the resolution of sub-10 nm, which is far from atomic resolution needed to directly discern the crystallinity. Instead we have compared the density change over an extended region using images obtained through coherent diffraction and phase retrieval. A simple data analysis to compare the obtained images with those from intact crystalline specimens, as described in the manuscript, is performed to distinguish the density reduction near the surface (10 ps & 40 ps, for instance), and we attributed this density reduction near the surface to be caused by melting.

To support the sensitivity of the density difference in characterizing regions with lower density, we provide the simulation result below. Images showing plane projected density of a nanosphere are provided for a solid sphere and a solid with 20% lower density value (for instance $\sim 7.7\%$ expansion in length effectively) in the spherical shell. The line plots through the center of the circle are shown to compare the projected density of the intact solid sphere and the expanded one with exterior having lower density due to the melting. In itself, the projected density image does show the change notably, as speculated by the referee as well. However, the comparison of the density from the surface melt sphere with the ideal sphere clearly distinguishes the region of reduced density. As shown in the attached images, the difference of the two projected density clearly visualize the melted region. This exact approach is taken in our manuscript.

[Comment 1-b]

b) The given fluence value seems odd. Fluence thresholds for particle modifications are frequently studied and have shown that even in a dense medium (such as a liquid) with stronger plasmon resonance about 10-15 mJ/cm² are needed to bring the particles close to the melting point (Boutopoulos, *Nanoscale* 2015, Plech *Phys. Rev. B* 2004, Katayama *Langmuir* 2014, Lombard, *Phys. Rev. Lett.* 2015, Hashimoto *J. Photochem. Photobiol. Rev.* 2012). Melting enthalpy has to be added here. Even if the given fluence is supposed to melt the particles completely it does not necessarily result in particle disintegration. Nevertheless, near-field induced ablation has been reported earlier to happen around the melting point, not necessarily in the gold liquid phase (ref. 22, which btw excludes the notion "for the first time" on top of page 5). So a meaningful report needs to prove the real processes and thresholds.

<Author reply>

We have revised the manuscript to indicate the incident laser fluence value, and an estimated value of the absorbed fluence. Details on how we have estimated the absorbed fluence is provided in the SI, where we used the nominal index of refraction to calculate the absorbance.

Most of all, we would like to mention that the main message of our work is not about extracting the exact value of the threshold fluence to melt a single Au particle. The laser fluence value is calibrated and the experiments were repeated several times to support our description. There can be a discrepancy while calculating the actual absorbed fluence by the Au nanosphere, where we have used the table value in calculating the absorbance and also considered possible fluctuation and peak pulse energy drift to provide the average fluence (~ 75 % of the nominal value, as described in the SI) and geometrical stretching (~ 80%) in calculating the effective absorbed fluence. Without considering all this, the incident laser fluence was 17uJ focused down to 50 um in diameter ($\sim 0.8 \cdot 870 \text{ mJ/cm}^2 \sim 700 \text{ mJ/cm}^2$).

Furthermore, we also think the threshold fluence that the referee has speculated from the listed research articles may not be relevant for a direct comparison. Here we are providing the result of the gold nanoparticles in vacuum and considered particle disintegration via

melting transitions, instead of specimens in solution and “ablation dominated” processes that the referee is broadly referring to. The discrepancy in the fluence is described in the cited text by E. Gamaly on ‘Femtosecond Laser-Matter Interactions’.

Further to support our value, we list some published results here also. The two-temperature molecular dynamics simulations on 50 nm free standing Au film by Zhigilei and colleagues have shown that irradiation of 200 fs laser pulse with $F = 13 \text{ mJ/cm}^2$ leads to melting of the film by $\sim 100 \text{ ps}$. Femtosecond electron diffraction measurements by J. R. D. Miller and colleagues confirmed this prediction by showing that a 20 nm Au film melts completely by 10 ps after the laser irradiation with $F = 11.9 \text{ \& } 13.7 \text{ mJ/cm}^2$ (Dwyer, Phil Trans R. Soc A. 2006, Dwyer, J. Mod. Opt. 2007). Systematic investigation of damage threshold was reported to support the range $\sim 10 \text{ mJ/cm}^2$ for 100 nm thin Au (S.-S. Wellserhoff et al., Appl. Phys. A. 1999).

Fig. 2b of Reference 22

Fig. 2 of our manuscript

Near field enhancements to assist the ablation process is reported in reference 22 indeed. Our work’s main emphasis is on directly unveiling, for the very first time, the particle melting and disintegration processes with real images of specimens at sub-10 nm spatial and sub-10 ps temporal resolution, which is a clearly different issue from that raised by the reviewer. To comment on it in more detail, reference 22 provides the particle morphology inferred from SAXS patterns at a few ns scale. On this time scale, one observes the end result referring to the eventual particle destruction, but the process of structural transformation is lost. Compared to Fig. 2b of reference 22, we provide the images of real specimens over the course of irreversible structural changes on a picosecond time scale (Fig. 2 of our manuscript). The image of eventual particle destruction (140 ps and later) observed by our experiment is similar to Fig. 2b of reference 22. With this, we are reassured that our observation on irreversible structural changes of gold nanoparticles at picosecond and nanometer spatio-temporal resolution via single-shot imaging is the first of its kind, and offers novel knowledge on unexplored regime as a truly revolutionary step.

[Comment 2-a]

2. A close inspection of the reconstructed projected density maps reveals some odd perceptance:

a) I assume that all data stem from single-shot excitation of individual particles (i. e. every particle can only be used once before both particle and support are destroyed). Can the authors confirm this? In that case it seems surprising that for instance delays of 60, 80 and 100 ps produce very similar damage patterns, which imply a time sequence on the very same particle. Did the authors sort a large number of data in order pick particularly those, which seem similar?

<Author reply>

All the images are taken from single shots. Each particle is used only once, and destroyed completely after the exposure (Please refer to the attached image with broken membrane windows above). The images at different delay times were obtained from independent particles. We obtained a few hundred independent images in total. Images are sorted for the same time delay, and they display similar patterns in morphology change with a small deviation. A couple of different images for the same delay time are displayed in SI Fig. 2, and the average behavior of the sample expansion is provided in SI Fig. 5 of the submitted version, which is now included in the main figure as suggested (Fig.3b in the revision).

Through the revised SI (Supplementary Figs.2&4), we now provide more images collected at the same time delay, which expects to accommodate better understanding of the process. Figures in the manuscript (Fig.3a, for instance) are prepared by selecting an image well representing others of the same delay time. We also revised the main manuscript by expanding the explanation on the data presentation to add the following text in page 4, “All the images were obtained from individual nanoparticles with very consistent 3D morphology, each of which was exposed to the fs IR laser only once (Methods). The reconstructed images were sorted out for the same delay time and tens of images were collected for each delay time (Supplementary Fig.2). From the collection of images, we chose one representing the average behavior for each delay time.”

[Comment 2-b]

b) the difference map in fig. 3. b is strange. Apparently in some spots the intensity increases by 70 %. How can that happen? Is the average value floating?

<Author reply>

The density difference map (in Fig.3) shows the local deformation of the sample departed from an ideal sphere of homogeneous density from an isotropic expansion. It is

designed to emphasize the local variation of the projected mass density from the ideal sphere of the same size. The total number of ions is preserved ($\sum \rho(x, y)_{sphere} = same$), and as such the value of projected mass density can be lower for an expanded nanoparticle. The degree of density difference per se does not indicate an actual amount of density increase nor loss from the intact solid Au nanoparticle. Instead it shows the degree of distortion from a sphere with homogeneous density (as expected from isotropic expansion). The value of 0.7 increase is not from the original particle. The ideal sphere provides the reference level of projected density at each time delay taking into account the radial expansion in accordance with the experiments.

In order to avoid any misleading interpretation of our results, we have removed the difference map images in Fig.3b. This, as explained earlier, was introduced to emphasize the local deviation of the projected density compared to an ideal sphere. To deliver the message more accurately while avoiding any potential misinterpretation, we have revised Fig. 3 to display the obtained images, which are the projected densities at different delay times, with the common colormap scale bar; the maximum projected density of the intact nanosphere gives one (bright yellow color). It traces the evolution of the actual projected density variation compared to that of the original nanoparticle more clearly. The image at 40 ps with subtle density variation is emphasized by comparing it with an ideal sphere, and this difference is shown without scale through the inset in the revised Fig.3a.

[Comment 3]

3. the discussion of a mechanistic excitation cascade on page 6 is clumsy and not precise. An assumption that momentum transfer from fast electrons to the ions might lead to anisotropic ion movement is pure speculation. Different views of local field effects, such as from emitted photoelectrons along the plasmon resonance is not discussed. Anyway, the simulation cannot give an answer to that as it only artificially imposes an asymmetry in energy deposition in eq. S4, not backed by any ab-initio knowledge. Also: "The void results from an enhanced local pressure caused by increased atomic displacements more amplified along the polarization direction." seems strange. Shouldn't it be a drop in pressure in the center that allows creating a void as usual in tensile disruption? Anyway, at this point comments on strain do not seem to be backed up by underlying (simulation) data, but rather serving as heuristic description.

<Author reply>

In response to the reviewer's comment, the whole paragraph has been completely rewritten. We have cited supporting references to all the sentences to avoid the impression that our interpretation is speculation. The mechanistic excitation cascade on page 6 is thoroughly rewritten to be more explicit with supporting evidence from published results as follows. "The void results from the release of the enhanced local pressure due to the increase of atomic

displacements more amplified along the polarization direction. The fs laser pulse excites localized surface plasmon, which induces enhanced electric fields at the nanoparticle surface and interior (Hartland et al. ACS Energy Lett. 2017, Varin et al. PRL 2012). This near-field enhancement is intense near the pole region of the sphere surface and decreased with radial distance inside the nanoparticle (Kundu Phys. Plasmas 2013, Varin et al. PRL 2012). As a result, a local area with more energetic electrons (i.e. a hot spot) is formed instantaneously, which eventually thermalizes and equilibrates with other electrons (Zheng et al. Nat. Commun. 2015). These transiently excited hot electrons create the reduced charge screening to weaken the interatomic bonding between Au ions in the local area [Daraszewicz et al. PRB 2013]. Several mechanisms can account for the reduced screening, including the excitation of conduction electrons, photoelectron emission via single- or multi-photon absorption, etc [Fennel et al. Rev. Mod. Phys. 2010]. The combination of the anisotropically weakened interatomic bonding and the energy transfer from the highly excited electrons to the lattice trigger ionic pressure accumulation, which is relieved primarily through the region with the weakened interatomic bonding. This process drives the anisotropic melting with the void formation as directly observed from our single-shot imaging experiment [Ivanov & Zhigilei PRB 2003, Vinet et al. PRB 1987].”

Summarizing the mechanistic excitation cascade elaborated in the revised manuscript, an IR laser pulse excites localized surface plasmon, which in turn anisotropically enhances the incident electric field on the nanoparticle surface and interior. The energetic electrons are generated at these hot spots with an enhanced electric field. This anisotropic excitation of electrons weakens the interatomic bonding through reduction in charge screening (Daraczewicz et al. 2013). Under this modified interatomic potential, ions quickly gain kinetic energy via high energy electron-to-phonon energy transfer, accelerating anisotropy in ionic kinetic energy and local pressure. The local pressure builds up at the hot spots until the lattice undergoes volume change. As the equation of state of solids gives the inverse relationship between pressure and volume (Vinet et al. 1987), the pressure build-up at the hot spot is released via local volume expansion, which eventually leads to void formation and disintegration of the gold nanosphere.

TTMD itself cannot provide ab initio level of interpretation on the involved physical process, and the related theory on ultrafast laser matter interaction is also under rapid development. However, the result from our TTMD simulations with the experimental observation as the boundary condition well follows the experimental results, and provide good

insight into the physical processes involved, helping to facilitate our understanding. Stimulated from our observation, we anticipate a more thorough theoretical investigation to eventually gain more accurate understanding on the ultrafast laser matter interaction.

[Comment 4]

4. Besides the nice data it is difficult to distill real new insight gained by the study by just watching single events. I guess, figure S4 would be the most quantitative outcome, worth being incorporated in the main text (while the dotted lines are only guide to the eye?)

<Author reply>

The results we provide are not from single events, but from a collection of multiple events with the representative images for the figures. This whole process is explained several times throughout this reply. The revised manuscript and SI state this repeatedly to avoid future misunderstanding. Following the referee's suggestion, we have included Fig. S4 in the main section of the revised manuscript as Fig.3b. The dotted lines are only a guide to the eye and we do not intend to make any serious interpretation on it.

REVIEWERS' COMMENTS:

Reviewer #1 (Remarks to the Author):

The revised submission partially answers my concerns and fills in some of the missing details about the experiment.

The "simulation" modelling is sheer speculation and has not improved at all in the revision. While a modest amount of speculation is useful in an experimental report, the authors appear to attach far too much weight to this part.

I will agree to publication if the authors will move lines 134-153 into the supplementary information. The remaining level of speculation would be acceptable to me.

I am happy to see that the other referee has similar opinions about the balance of the work.

Reviewer #2 (Remarks to the Author):

The authors have sufficiently responded to the criticism presented by the reviewers. The manuscript has considerably improved in clarity and may be published.

I still recommend rethinking two aspects that were discussed:

1. The authors have added that the incoming fluence was indeed 700 mJ/cm^2 , which (even with reduced cross section in vacuum) is high enough to explain the fragmentation. On the other hand, speaking of an "absorbed fluence" of 14 mJ/cm^2 seems not helpful. It would be better to name the absorbed energy (say in fJ).

2. Both reviewers were wondering about the asymmetry of void formation in different (independent) events, which, in Fig. 2 of the supplementary information, always starts on the top part rather than the bottom part of the particle (parallel to laser polarization). Have the images been rotated or is there another explanation of this symmetry breaking (such as slightly tilted X-ray or laser path relative to the substrate or relative to each other)?

Trusting that the authors can comment on that I don't need to see the manuscript again.

Response to reviewers' comments

[Reviewer #1]

The revised submission partially answers my concerns and fills in some of the missing details about the experiment.

The "simulation" modelling is sheer speculation and has not improved at all in the revision. While a modest amount of speculation is useful in an experimental report, the authors appear to attach far too much weight to this part. I will agree to publication if the authors will move lines 134-153 into the supplementary information. The remaining level of speculation would be acceptable to me. I am happy to see that the other referee has similar opinions about the balance of the work

<Author reply>

Following the referee's suggestion, we have moved the main text from line 134 to 153 in the manuscript to the Supplementary Information as Supplementary Discussion. Instead, we included short description on the TTMD results in the revision. As it is very important to provide the readers with relevant information on the results, a brief description on the MD results remains in the main text. No speculation is attempted for this and only essential information is used.

[Reviewer #2]

The authors have sufficiently responded to the criticism presented by the reviewers. The manuscript has considerably improved in clarity and may be published. I still recommend rethinking two aspects that were discussed:

1. The authors have added that the incoming fluence was indeed 700 mJ/cm^2 , which (even with reduced cross section in vacuum) is high enough to explain the fragmentation. On the other hand, speaking of an "absorbed fluence" of 14 mJ/cm^2 seems not helpful. It would be better to name the absorbed energy (say in fJ).

2. Both reviewers were wondering about the asymmetry of void formation in different (independent) events, which, in Fig. 2 of the supplementary information, always starts on the top part rather than the bottom part of the particle (parallel to laser polarization). Have the images been rotated or is there another explanation of this symmetry breaking (such as slightly tilted X-ray or laser path relative to the substrate or relative to each other)?

Trusting that the authors can comment on that I don't need to see the manuscript again.

<Author reply>

To comment 1: Thanks for the suggestion. We have removed the estimation on the absorbed fluence following the recommendation. Absorbed energy density is estimated in the revision as suggested.

To comment 2: Both with no physically meaningful distinction between up or down in this experiment and with the same diffraction pattern for the specimen with 180-degree rotation along the axis normal to the image plane, images with 180-degree rotation are not distinguished. As such, we have aligned the images to show the void from the upper hemisphere by rotating images by 180 degree, if necessary. We have stated about this explicitly in the manuscript now.